# Cancer Testis Antigens and Immunotherapy: Where Do We Stand in the Targeting of PRAME?

**DOI:** 10.3390/cancers11070984

**Published:** 2019-07-15

**Authors:** Ghaneya Al-Khadairi, Julie Decock

**Affiliations:** 1College of Health and Life Sciences (CHLS), Hamad Bin Khalifa University (HBKU), Qatar Foundation (QF), Doha P.O. Box 34110, Qatar; 2Cancer Research Center, Qatar Biomedical Research Institute (QBRI), Hamad Bin Khalifa University (HBKU), Qatar Foundation (QF), Doha P.O. Box 34110, Qatar

**Keywords:** PRAME, cancer testis antigen, cancer vaccine, adoptive T cell therapy, immunotherapy, antibody

## Abstract

PRAME or PReferentially expressed Antigen in Melanoma is a testis-selective cancer testis antigen (CTA) with restricted expression in somatic tissues and re-expression in various cancers. It is one of the most widely studied CTAs and has been associated with the outcome and risk of metastasis. Although little is known about its pathophysiological function, PRAME has gained interest as a candidate target for immunotherapy. This review provides an update on our knowledge on PRAME expression and function in healthy and malignant cells and the current immunotherapeutic strategies targeting PRAME with their specific challenges and opportunities. We also highlight some of the features that position PRAME as a unique cancer testis antigen to target.

## 1. Introduction

Conventional cancer care, including surgery, radiotherapy, and chemotherapy, considerably prolongs the survival of patients; however, a significant proportion of cancer patients do not respond to standard systemic therapy or relapse after treatment. The need for alternative, well-tolerated first-line treatments has been partially met by the development of targeted therapies involving a number of novel therapeutic drugs that target specific molecules. For example, antibody-based targeting of the Epidermal Growth Factor Receptor (EGFR), Human Epidermal Growth Factor Receptor 2 (Her2), and Cluster of Differentiation 20 (CD20) are currently used as standard targeted treatments in lymphoma, breast, colorectal, and head and neck cancers [1,2,3]. Despite initial success with targeted therapy, many patients develop resistance and do not benefit from the treatment in the long term. Thus, the need for alternative, well-tolerated cancer treatment strategies that can induce durable clinical responses remains. Recently, cancer immunotherapy has been acclaimed as the long-awaited breakthrough in cancer treatment, providing durable and potent clinical efficacy in a range of cancer types. The revolutionary nature of immunotherapy is evident from the success stories using immune checkpoint inhibitors. Checkpoint blockade using antibodies against Programmed Death 1 (PD-1) or its ligand, Programmed Death Ligand 1 (PD-L1), and Cytotoxic T-Lymphocyte-Associated Antigen 4 (CTLA-4) has greatly enhanced the anti-tumor immune response, improving clinical outcomes and leading to durable responses in numerous cancers [4]. Furthermore, cellular immunotherapy using ex vivo expanded circulating antigen-specific T cells or tumor infiltrating lymphocytes (TILs) and engineered T cells has gained interest in the field, as it does not rely on the patient to actively generate large numbers of antigen-specific immune cells and has shown a significant increase in durable complete response rates across malignancies [5]. Treatment with autologous expanded TILs has resulted in durable complete responses in 22% of metastatic melanoma patients at up to 109 months of follow-up [6]. Likewise, durable responses have been obtained using autologous T lymphocytes that are engineered to target intracellular antigens through T cell receptors (TCRs) or cell surface antigens through Chimeric Antigen Receptors (CARs), followed by ex vivo expansion into sufficient numbers and selection for avidity and activity prior to infusion into the patient. The success of engineered T cell therapy is apparent from clinical trials with CD19-CAR T cells, where 61–93% of patients with B cell acute lymphoblastic leukemia achieved a complete response at up to 18 months, and from the trials with NY-ESO-1 specific TCR T cells where objective responses were observed in 55% of melanoma patients with a 5 year survival rate of 33% [7,8,9]. However, the clinical utility of adoptive T cell therapy is challenged by the identification of candidate target antigens with limited on-target and/or off target toxicity. In this respect, cancer testis antigens (CTAs) have been considered promising targets for adoptive T cell therapy thanks to their restricted expression in somatic normal tissues, re-expression in many cancer types, and immunogenic nature [10]. Approximately half of all CTAs are encoded on the X-chromosome, aptly termed CT-X antigens. Together they make up 10% of all genes on the chromosome and are induced by demethylation during spermatogenesis [11]. CTA re-expression in tumors has been linked with worse clinical outcome in various cancers [12]. Since the majority of CTAs are intracellular proteins, efforts to develop CTA-based cellular immunotherapy have focused on the isolation and expansion of circulating CTA-specific T cells and on the engineering of antigen-specific TCR T cells. Among all candidate CTAs, MAGE-A3, NY-ESO-1, and PRAME have shown great potential as prognostic biomarkers and immunotherapeutic targets [9,13]. This review focuses on PReferentially expressed Antigen in Melanoma, or PRAME, in relation to its expression profile, biological functions, and our current knowledge on PRAME-tumor immunology and immunotherapy. Targeting PRAME presents an attractive approach for CTA-based immunotherapy, as it displays few unique features in comparison to other cancer testis antigens, which are highlighted in this review.

## 2. PRAME Expression in Healthy and Neoplastic Tissues

PRAME, also known as CT130, MAPE (Melanoma Antigen Preferentially Expressed in tumors) or OIP4 (OPA-Interacting Protein 4), is encoded over a region of approximately 12 kilobases on chromosome 22 at locus 22q11.22. It was first identified as a tumor antigen that could be recognized by human leukocyte antigen (HLA)-A*24 restricted cytotoxic T lymphocytes in metastatic cutaneous melanoma [14]. PRAME is better classified as a testis-selective rather than a testis-restricted CTA, as low RNA expression has been observed in endometrial, ovarian, and adrenal gland tissues in addition to the testes [15]. PRAME expression is primarily upregulated by DNA demethylation, at least in part in conjunction with binding of the transcription factor MZF1, with recent evidence suggesting that aberrant PRAME expression can also be induced by miRNA-211 downregulation [16,17]. PRAME is re-expressed in a wide range of cancers, including melanoma [18,19], renal cell cancer [20], non-small cell lung cancer (NSCLC) [21], neuroblastoma [22], breast cancer [23,24], multiple myeloma [25,26], acute leukemia [27,28], chronic myeloid leukemia [25,29], multiple sarcoma subtypes [30,31], and primary and metastatic uveal melanoma [32,33]. Given the widespread expression of PRAME across different cancers, it is important to consider its heterogeneity of expression, as this may limit the PRAME-specific therapeutic response. Cancer testis antigens such as the well-studied NY-ESO-1 are expressed at different levels within a single tumor; hence, it is not unlikely that PRAME expression might also display intra-tumor heterogeneity [34,35]. In accordance, differences in PRAME expression were found among 14 single cell clones derived from a single cutaneous melanoma lesion, suggesting that the presence of low expressing cell clones may allow cancer cells to escape PRAME-targeted immunotherapy [34]. One approach to overcome treatment resistance by these distinct clones would be to induce PRAME expression for targeting purposes. Indeed, several studies have demonstrated that treatment with the demethylating agent 5′-aza-2′-deoxycytidine can successfully induce the expression of CTAs, including PRAME [36,37,38]. Moreover, PRAME expression has been associated with poor prognosis in solid cancers including breast cancer [23,24], sarcoma [39,40,41], head and neck cancer [42], medulloblastoma [43], uveal melanoma [32,33], and neuroblastoma [22]. In contrast, a number of reports have suggested that increased PRAME expression is associated with a favorable outcome and treatment response in various hematological malignancies [44,45,46,47,48]. Its prognostic significance is further corroborated by its value for monitoring minimal residual disease in patients with acute leukemia [44,45,49]. Together these findings suggest that PRAME could be a valuable target for immunotherapy alongside its fellow cancer testis antigen NY-ESO-1.

## 3. Biological Functions of PRAME in Cancer

The cellular and molecular functions of PRAME in normal and neoplastic cells remain largely unclear and may differ depending on tissue-specificity and/or subcellular localization of PRAME. PRAME was identified as a dominant repressor of retinoic acid (RA) signaling, thereby inhibiting differentiation, proliferation arrest, and apoptosis [50]. Its overexpression was found to inhibit myeloid cell differentiation of leukemic progenitor cells [51]. In line with this, leukemic CD34+ progenitor cells of patients with advanced phase CML express high levels of PRAME in comparison to the CD34+ cells of healthy individuals [52]. Of note, the silencing of *PRAME* could restore sensitivity to retinoic acid treatment in RA-resistant melanoma cell lines, thereby reducing cancer cell proliferation by de-repressing the expression of RAR target genes [50]. However, whether the increased expression of PRAME could be linked to resistance to all-trans retinoic acid (ATRA) therapies in patients with acute promyelocytic leukemia (APL) remains to be determined [53]. In this respect, PRAME targeting could offer a unique opportunity, not bestowed to other CTAs, to enhance the treatment response to ATRA treatment.

In recent years, however, it has become apparent that PRAME could promote tumor development and progression via different mechanisms. Its nuclear localization has been implicated in transcriptional regulation, thereby possibly driving the expression of genes that are involved in tumor promotion. More specifically, analysis of the PRAME interactome revealed that PRAME facilitates the recruitment of Cullin2 ubiquitin ligases to the EKC/KEOPS complex in the nucleus, which is involved in telomere maintenance, transcriptional regulation, and threonylcarbamoyladenosine (t6A) modification of transfer RNAs (tRNA) for accurate decoding of A-starting mRNA codons [54,55,56].

Furthermore, the increase in PRAME expression with more advanced disease and a higher risk of metastasis suggests that PRAME plays a role in the acquisition of various cancer hallmarks, including replicative immortality or stemness, and invasion and metastasis [32,33,39,40,41,57,58]. Thus, we recently explored the biological mechanisms by which PRAME could facilitate cancer cell motility as one of the major biological processes involved in regulating metastatic behavior [59]. We demonstrated that PRAME increased the migratory and invasive behavior of triple negative breast cancer cells in vitro by promoting the epithelial-to-mesenchymal transition (EMT).

In addition to supporting tumor cell features, PRAME has been implicated in the regulation of the immune response. Homology sequence analyses revealed that PRAME contains 21.8% (iso)leucine residues, and hence belongs to the leucine-rich repeat (LRR) family of proteins [25]. Furthermore, structural analyses indicated similarities with Toll-like receptors (TLR3 and TLR4) that play an important role in the recognition of pathogen-associated molecular patterns (PAMPs) during antimicrobial immune responses. In a follow-up study, the authors demonstrated that PRAME is upregulated in response to bacterial PAMPs and IFN-γ, followed by translocation to the Golgi network where it co-localizes with the Elongin/Cullin E3 ubiquitin ligase complex [60]. This suggests that cytoplasmic PRAME might be involved in protein ubiquitylation in response to pro-inflammatory stimuli. How PAMP/IFN-γ treatment induces PRAME expression remains unknown, but this could involve the activation and binding of transcription factors to the proximal promoter of *PRAME*, including nuclear factor kappa B (NFĸB), interferon regulatory transcription factors (IRFs), and signal transducers and activators of transcription (STATs) [61]. PRAME has the ability to bind to bacterial OPA proteins, further supporting a role for PRAME in the innate immune response [62]. This raises the question of whether PRAME could be involved in regulating the adaptive anti-tumor immune response, corroborating its value as an immunotherapeutic target in comparison to other CTAs. Moreover, given the association of PRAME expression with favorable outcomes in hematological diseases, directly targeting PRAME by small molecule inhibitors or gene silencing might not be as beneficial as compared with cytotoxic targeting of the tumor cells expressing PRAME.

## 4. PRAME-Targeted Immunotherapy

The restricted re-expression of PRAME across malignancies renders it a candidate target for cancer treatment and has sparked numerous studies on the development and evaluation of immune-based interventions targeting PRAME. A variety of immunotherapy strategies are currently developed with the common goal of eliciting durable and potent anti-tumor immune responses. To date, efforts in PRAME immune-based targeting have focused on cancer vaccines and adoptive T cell therapies, as described below. Ongoing clinical trials of cellular immunotherapies targeting PRAME are summarized in Table 1.

### 4.1. Cancer Vaccines

Clinical trials on PRAME cancer vaccines have reported no major toxicities in patients with various solid cancers; however, thus far, the therapeutic benefits have been rather limited, hence discouraging the further development of PRAME vaccines [63,64]. A first hint at a possible explanation for these suboptimal results can be traced back to the lack of CD8+ cytotoxic T cell responses in the early preclinical study of the acellular recPRAME+AS15 vaccine containing the recombinant PRAME protein (recPRAME) and the AS15 immunostimulant (AS15) [65]. Whereas PRAME-specific humoral and CD4+ and CD8+ responses were observed in HLA-A02.01/HLA-DR1 transgenic mice, no CD8+ responses were detected in nonhuman primates (cynomolgus monkeys). These findings were corroborated in phase I dose escalation studies in human metastatic melanoma (NCT01149343) and non-small cell lung cancer (NCT01159964), demonstrating PRAME-specific antibodies and CD4+, but not CD8+, T cell responses [63,66]. This phenomenon is not restricted to PRAME but has also been observed when targeting other tumor-associated, but not tumor-selective, antigens, and it might be related to the negative selection of high affinity T cell responses in the thymus. Indeed, given the low expression of PRAME in endometrial, ovarian, and adrenal gland tissues, it is likely that high affinity PRAME-reactive cytotoxic T cells undergo negative thymic selection in order to avoid autoimmune responses against those normal tissues where PRAME expression could be regarded as a self-antigen. Further evaluation of this vaccine in a phase II NSCLC study (NCT01853878) was terminated prematurely, as the benefit of CTA cancer vaccines in NSCLC was questioned given the strong immunosuppressive environment of NSCLC tumors and the lack of improvement in clinical outcomes of a phase III randomized MAGE-A3 vaccination study [67]. In contrast, promising results have been obtained using a cancer vaccine targeting PRAME together with prostate-specific membrane antigen (PSMA). Treatment with this multi-tumor associated antigen (TAA) vaccine resulted in the expansion of the antigen-specific CD8+ T cell population with stable disease (6 to 18+ months) in four out of ten patients with refractory prostate cancer and in two out of two patients with metastatic renal clear cell carcinoma [64]. Another multi-TAA peptide vaccine approach directed against PRAME, NY-ESO-1, MAGE-A3, and WT-1 in combination with 5′-aza-2′-deoxycytidine is currently in phase I trial in patients with myelodysplastic syndrome and acute myeloid leukemia (NCT02750995). In comparison to peptide cancer vaccines, cellular vaccine therapy targeting PRAME is a less active and well-developed field. Currently, only one clinical trial is exploring the safety and efficacy of WT1/PRAME-loaded dendritic cells in patients with acute myeloid leukemia in remission (NCT02405338). Potentially, cellular vaccines could confer a greater efficacy through the specific priming of pre-existing antigen-specific T cell clones.

### 4.2. Adoptive T Cell Therapy

Cellular immunotherapy with PRAME-specific T cells has largely been explored using two different approaches involving either the isolation of pre-existing circulating PRAME-specific T cells for ex vivo expansion and selection, or the genetic engineering of high-affinity PRAME-specific TCR T cells. As aforementioned, PRAME was originally identified through the detection of specific tumor-reactive HLA-A*24–restricted T cell clones in a metastatic melanoma patient [14]. Analysis of proteasome cleavage of PRAME polypeptides in conjunction with in silico epitope prediction identified four additional HLA-A*0201–restricted epitopes: PRA100-108 (VLDGLDVLL), PRA142-151 (SLYSFPEPEA), PRA300-309 (ALYVDSLFFL), and PRA425-433 (SLLQHLIGL) [68].

Following the identification of these candidate epitopes, patients’ blood samples were screened to evaluate each epitope’s ability to induce specific T cells, thereby providing a rationale for cellular immunotherapy using ex vivo expansion of circulating autologous antigen-specific T cells. In vitro priming of isolated peripheral blood mononuclear cells with peptide-pulsed autologous antigen presenting cells demonstrated the presence of PRA100-108 and PRA300-309-specific T cells in 36% of melanoma and 70% of AML cancer patients, respectively [48,69]. Furthermore, the ex vivo expanded PRA100-108 and PRA300-309-specific T cells exhibited cytotoxic activity against autologous chronic myeloid leukemia (CML) blasts and cancer cell lines of various histologic origins [69,70]. However, these T cell clones have a rather low affinity, possibly due to the aforementioned thymic negative selection of high-affinity clones, which limits their clinical application. In order to increase the avidity, polyclonal multivalent cytotoxic lymphocytes were generated from healthy donors and leukemia patients using a peptide library of 125 synthetic pentadecapeptides [70]. Of note, a large proportion of the cells showed activity against a novel HLA-A*02–restricted epitope (NLTHVLYPV, PRA435-443) that is expressed in HLA-A*02+ PRAME+ tumor cell lines and primary leukemic blasts. These high-avidity T cells not only showed activity against leukemic blasts but also against leukemic progenitor cells without affecting normal hematopoietic precursors, thereby expanding the pool of neoplastic cells that can be targeted by PRAME adoptive T cell therapy. In parallel, another research group isolated PRAME-specific HLA-A2 restricted T-cell clones with high specificity for PRAME (HSS1 and HSS3) as a result of a graft-versus-host immune response to an HLA-A2 mismatched stem cell transplantation [71]. These T cells were able to recognize various PRAME-expressing cell lines derived from melanoma, acute myeloid leukemia, colon carcinoma, cervix carcinoma, and lung and breast cancer, whereas acute lymphoblastic leukemia cells were only recognized after CD40L-activation of the T cell clones. To date, numerous ongoing clinical trials are studying the safety and efficacy of targeting PRAME using ex vivo expanded multi-TAA-specific T cells in hematological malignancies (NCT02291848, NCT02475707, NCT02494167 and NCT02203903), neuro-oncologic disease (NCT03652545), rhabdomyosarcoma (NCT02239861), pancreatic cancer (NCT03192462), breast cancer (NCT03093350), and a range of high-risk solid tumors (NCT02789228).

In comparison to the research efforts on cellular therapy using pre-existing autologous PRAME-specific T cells, there is very limited published information on PRAME genetically modified T cell therapy. Notwithstanding, adoptive therapy using TCR transduced T cells has shown encouraging results in generating durable responses in selected cancer types, resulting in a surge of clinical trials targeting various tumor-associated antigens, including the cancer testis antigens NY-ESO-1. For instance, NY-ESO-1 specific TCR-T cells are currently being evaluated in numerous clinical trials following encouraging results with improved clinical response rates and prolonged survival of patients with metastatic melanoma, synovial sarcoma, and multiple myeloma [9]. A recent study reported that the transduction of T cells with a high affinity PRAME TCR and an inducible caspase-9 safety switch (BPX-701) eliminated the genetically modified T cells in case of detrimental toxicity [43]. The BPX-701 TCR T cells were capable of eradicating both medulloblastoma cell lines and primary medulloblastoma cells in vitro. Moreover, the transduced T cells induced tumor regression in an orthotopic medulloblastoma mouse model without excess neurological toxicity and were successfully eradicated by the suicide gene. These BPX-701 T cells are currently under evaluation in a phase I clinical trial in patients with relapsed AML, refractory myelodysplastic syndrome (MDS), and metastatic uveal melanoma (NCT02743611). A second clinical trial with a different type of PRAME TCR T cells (MDG1011) will evaluate their safety and efficacy in patients with AML, MDS, and multiple myeloma (NCT03503968).

It is important to note that the efficacy of adoptive T cell therapy is not only determined by the avidity of the T cell clones but also by the expression level of the target throughout the tumor. As mentioned earlier, tumors often show heterogeneous tumor-associated antigen expression patterns so that distinct tumor cell clones with absent or relatively low target expression might escape T cell killing. In order to tackle this, it would be beneficial to induce re-expression of the target in all tumor cells using demethylating agents (DMA), histone deacetylase inhibitors (HDACi), or a combination of both. These epigenetic modulators have been successfully used to upregulate tumor PRAME expression, resulting in an increase in cytolytic activity of ex vivo expanded autologous T cells [72,73]. The approach to combine demethylating agents with PRAME T cell transfer is currently under phase I clinical trial using 5′-aza-2′-deoxycytidin and multi-TAA TCR T cells targeting up to five antigens (NY-ESO-1, MAGEA4, PRAME, Survivin and SSX) in Hodgkin and non-Hodgkin lymphoma (NCT01333046).

## 5. Current Challenges and Opportunities for PRAME-Targeted Immunotherapy

PRAME has emerged as a potential candidate target for immunotherapy, and we are currently awaiting the results from numerous clinical trials evaluating PRAME-specific cancer vaccines and adoptive T cell therapies. Although preclinical studies have been promising, we should be vigilant and anticipate, as for any novel treatment approach, that PRAME-targeted immunotherapy might face challenges in the clinic. As depicted in Figure 1, every challenge can be seen as a novel opportunity for treatment development and improvement. PRAME expression has traditionally been observed in the nucleus and cytoplasm of tumor cells; therefore, immune-based strategies designed towards intracellular targets have been the treatment of choice, including acellular and cellular cancer vaccines and adoptive T cell therapy with autologous or TCR-engineered T cells. Recent evidence suggests that PRAME could be targeted as a membrane-bound protein using a specific monoclonal antibody (mAb) [74] or TCR mimic antibody (TCRm) [75,76], which also opens up the possibility for CAR-based T cell therapy. Overall, the efficacy of targeted immunotherapy may be hampered due to intra-tumor heterogeneity, loss or downregulation of HLA class I molecules as well as expression of immune checkpoint molecules such as PD-L1. Countermeasures for these include demethylation agents and histone deacetylase inhibitors to increase target expression, and MEK inhibitors and NK cell therapy to induce HLA expression. Immune checkpoint blockade could be used in combination with any of the PRAME-targeting approaches.

As a first challenge, the heterogeneity of PRAME expression could likely present a bottleneck for the efficient targeting and complete eradication of tumor cells. Firstly, PRAME expression shows a wide range of inter-tumor heterogeneity among patients carrying tumors of the same tissue origin, thereby limiting the overall response rate. This is of particular importance for breast cancer where only 27% of tumors display PRAME expression [19]. Secondly, tumor types that are defined as PRAME-positive might display significant intra-tumoral heterogeneity, leading to treatment resistance due to the low expressing cell clones. This dose-dependent immune response has been demonstrated in the context of PRA100-108 specific T cell activation in HLA-A*0201+ acute lymphocytic leukemia (ALL) and melanoma cell lines. While cytolytic activity was observed in melanoma cell lines, no activation of the T cell clones could be detected in the presence of ALL cell lines, which was postulated to be most probably linked to the lower expression of *PRAME* in the latter [69]. Moreover, an association between PRAME expression levels and multi-epitope PRAME-specific CD8+ T cell responses has been reported in leukemia [77]. In the same study, it was shown that leukemia patients naturally have higher frequencies of PRAME-specific precursor T cells compared to healthy controls. It remains to be determined whether this is linked to the ability of pre-existing PRAME-specific T cells to target leukemic progenitor cells in addition to leukemic blasts [70]. Together, these findings suggest that there is possibly a recognition threshold for PRAME expression in order to induce cytotoxic T cell responses. This is where epigenetic modulators, such as DMAs and HDACis, come into play to upregulate PRAME expression. However, caution is warranted as the impact of re-expression of a tumor-associated antigen in normal tissues remains unknown and epigenetic agents may alter immune cell function. Moreover, the upregulation of PRAME in low-expressing cancer cell clones will likely increase their metastatic potential via PRAME-mediated EMT; hence, the optimal dosage and therapeutic window of opportunity will need to be carefully considered when administering combination treatment of demethylation drugs and PRAME-specific immunotherapy.

Although increasing target expression represents one approach to improve the efficacy of targeted immunotherapy, positive tumor cells can still evade the anti-tumor immune response through several mechanisms. For example, dedifferentiated liposarcoma and leiomyosarcoma tumors with positive PRAME expression have been found to express reduced levels of antigen presentation molecules, such as HLA-B, HLA-C, beta-2 microglobulin (B2M), latent membrane proteins 2 and 7 (LMP2, LMP7), and transporter associated with antigen processing 2 (TAP2) [30]. This suggests that PRAME positive tumors can elicit an immune response and therefore downregulate the antigen presentation machinery in order to avoid immune recognition. Hence, it might be possible to improve the efficacy of PRAME-targeted immunotherapy using agents that can increase MHC class I expression, such as DNA demethylating drugs, HDAC inhibitors, inducers of the tumor suppressor Fhit, and MEK inhibitors [78]. Interestingly, combinatorial cellular immunotherapy using PRAME-specific T cells with NK cell-based therapy could also possibly increase the therapeutic response. Co-culture of activated NK cells with neuroblastoma cell lines demonstrated that the tumor cells could upregulate the expression of MHC class I molecules through NK contact-dependent production of IFN-γ, resulting in increased PRAME-specific T cell killing of the tumor cells [79]. On the other hand, PRAME expression has been observed in association with reduced PD-L1 expression in dedifferentiated liposarcoma, which appears to be contraindicative of the notion that PRAME could facilitate immune evasion. The molecular mechanisms underlying this association are currently unknown; however, numerous studies have reported on the regulation of the tumor expression of PD-L1. For example, PD-L1 expression can be induced by c-Myc [80], so it remains to be determined whether an increase in PRAME expression, alone or in conjunction with changes in c-Myc expression, could negatively regulate PD-L1 surface expression. Moreover, a recent study demonstrated that PD-L1 stability is regulated in a cell-cycle-dependent manner via a Cullin E3 ligase [81], which has been found to co-localize with PRAME in the Golgi network as previously mentioned. Nevertheless, whether PRAME is directly linked to PD-L1 and/or PD-1 expression, and hence anti-tumor immunity and response to the PD-L1/PD-1 blockade, remains unknown and warrants further investigation. Interestingly, PD-L1 expression is commonly not observed on blast cells from patients with AML, providing a potential mechanistic link for a PRAME-associated favorable outcome in these patients. Furthermore, PRAME-expressing tumors, such as advanced soft-tissue sarcomas, including dedifferentiated liposarcomas, exhibit a low response rate of 18% to anti PD-1 treatment (NCT02301039) [30,82]. Besides reduced PD-L1 tumor expression, many additional factors could contribute to resistance to anti-PD-1 therapy, including low numbers of exhausted high-avidity T cells, upregulation of other immune checkpoint molecules and receptors such as VISTA and TIM-3, loss of IFN-γ response elements, and macrophage-sequestration of the therapeutic anti-PD1 antibody [83]. Indeed, it is possible that pre-existing PRAME-specific T cells are at least partially anergic, since vaccination of patients with PRAME-specific T cells at baseline did not induce T cell expansion to the same extent as vaccination of patients without pre-existing PRAME specific T cells [64]. Hence, the question arises as to whether the efficacy of PRAME adoptive T cell therapy could be enhanced by treatment with immune checkpoint inhibitors. To the best of our knowledge, there are currently no published reports or active/recruiting clinical trials investigating this therapeutic approach.

Interestingly, antibody-based targeting of PRAME has emerged as a novel approach, which could offer a new avenue to induce potent anti-tumor responses. PRAME was, until recently, not considered a candidate target for antibody therapy, as monoclonal antibodies exclusively bind to the cell surface and extracellular antigens, while cancer testis antigens are intracellular antigens. However, a research group from the Memorial Sloan Kettering Cancer Center developed a TCR mimic human antibody that can recognize the PRA300–309 peptide in a complex with HLA-A2 [75]. The afucosylated Fc form of the Pr20 antibody was able to bind to HLA-A2+ PRAME+ leukemia and lymphoma cells in vitro and to induce antibody dependent cellular cytotoxicity (ADCC) in a dose-dependent manner. The authors reported that the TCR mimic required tumor cells to induce expression of the immunoproteasome in order to correctly process the PRA300–309 peptide, as demonstrated by the lack of Pr20 binding and ADCC in several HLA-A2+ PRAME+ melanoma cell lines. In a follow-up study, the authors demonstrated that antibody-dependent phagocytosis of PRAME-expressing cancer cells could be enhanced using a combinatorial treatment of PRAME TCRm mAbs with CD47 blockade, thereby releasing the inhibition of macrophage phagocytosis [76]. Thus, monoclonal antibodies targeting PRAME could be an additional viable treatment option for hematological as well as solid cancers in combination with IFN-γ mediated upregulation of the immunoproteasome, which renders PRAME a preferable CTA to target. This also opens up opportunities to enhance the anti-tumor response using antibody–drug conjugates or radio-immunoconjugates.

Although PRAME is widely believed to be an intracellular antigen, one study detected a membrane-bound form of PRAME in chronic lymphocytic leukemia and mantle cell lymphoma [84]. This result sparked the development of the polyclonal antibody Membrane associated PRAME Antibody 1 (MPA1) against the predicted extracellular PRA310-331 peptide [74]. The authors demonstrated specific binding of MPA1 to PRAME+ liquid and solid cancer cell lines, with increased binding upon demethylation treatment. However, the specificity of the MPA1 antibody was not evaluated in engineered cell line models to allow the evaluation of antibody binding in a single cell line before and after the manipulation of PRAME expression (knockout or overexpression). As a proof-of-principle model for in vivo targeting of PRAME using antibodies, the authors reported highly specific tumor uptake of radiolabeled MPA1 in xenograft mice. In order to limit cross-reactivity, they are currently developing a monoclonal antibody against the PRA300–309 peptide. The identification of a membrane-bound form of PRAME further opens up the possibility to target membrane-bound PRAME-expressing tumor cells with CAR T cell therapy, expanding the spectrum of tools for PRAME-targeted immunotherapy in comparison with the available approaches to target other CTAs.

## 6. Conclusions

Given the widespread re-expression of PRAME among malignancies and the novel developments in the immunotherapy field, we believe that there is great potential to target PRAME using a variety of immune-based interventions, either as single modality or as a combinatorial treatment. We discussed some of the major challenges for PRAME-based immunotherapy that need to be addressed, including the intra- and inter-tumor heterogeneous expression of PRAME and the potential role of PRAME in the upregulation of immune evasion mechanisms. Furthermore, we described the current countermeasures and innovative technologies that are being exploited to improve the potency of the PRAME-specific anti-tumor response.

Among all cancer testis antigens, PRAME is in a unique position as it is not only widely expressed in various cancers but also elicits specific cellular immune responses against numerous neoplastic cells as well as against leukemia progenitor cells, plays a role in the innate immune response and possibly also the adaptive anti-tumor immune response, could enhance the ATRA treatment response as a combinational therapy, and could be targeted using a wider range of approaches including antibody- and CAR-based therapies.

## Figures and Tables

**Figure 1 cancers-11-00984-f001:**
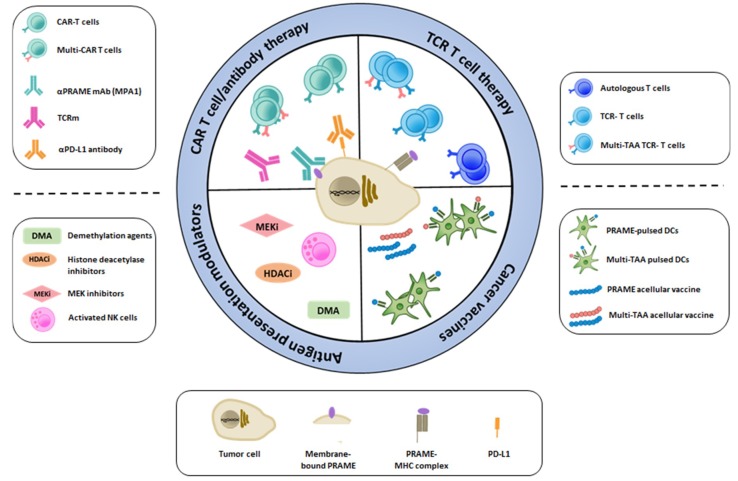
Immunotherapy approaches to target specific tumor-associated antigens such as PRAME.

**Table 1 cancers-11-00984-t001:** Ongoing clinical trials of cellular PRAME immunotherapy.

NCT Number	Other ID	Intervention	Disease
NCT01333046	H-27471, ACTAL	Multi TAA T cells (NY-ESO-1, MAGEA4, PRAME, Survivin and SSX), and Azacitidine	Hodgkin lymphoma, non-Hodgkin lymphoma, Hodgkin disease
NCT02203903	RESOLVE	Multi TAA T cells (WT1, PRAME, and Survivin)	Relapsed/refractory hematopoietic malignancies (ALL, AML, CML, MDS)
NCT02239861	H-35425, TACTASOM	Multi TAA T cells (NY-ESO-1, MAGEA4, PRAME, Survivin, and SSX)	Rhabdomyosarcoma
NCT02291848	H-35626, TACTAM	Multi TAA T cells (NY-ESO-1, MAGEA4, PRAME, Survivin, and SSX)	Multiple myeloma
NCT02475707	H-37042, STELLA	Multi TAA T cells (WT1, PRAME, and Survivin)	Leukemia, ALL
NCT02494167	H-36346, ADSPAM	Multi TAA T cells (WT1, NY-ESO-1, PRAME, and Survivin)	AML, MDS
NCT02743611	BP-011	BPX-701 and Rimiducid	AML, MDS, uveal melanoma
NCT02789228	7497	Multi TAA T cells (WT1, PRAME, and/or Survivin)	Solid tumors
NCT03093350	H-39209, TACTIC	Multi TAA T cells (NY-ESO-1, MAGEA4, PRAME, Survivin, and SSX2)	Breast cancer
NCT03192462	H-40378, TACTOPS	Multi TAA T cells (NY-ESO-1, MAGEA4, PRAME, Survivin, and SSX2)	Pancreatic cancer
NCT03503968	CD-TCR-001	MDG1011	High risk myeloid and lymphoid neoplasms
NCT03652545	REMIND	Multi TAA T cells (WT1, PRAME, and/or Survivin)	Brain tumor

TAA, tumor associated antigen; ALL, acute lymphocytic leukemia; AML, acute myeloid leukemia; BPX-701, PRAME-TCR T cells with iCasp9; CML, chronic myeloid leukemia; MDG1011, PRAME-TCR T cells; MDS, myelodysplastic syndromes; Rimiducid, homodimerizing tacrolimus analogue.

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
