# Peer review of "Cancer Testis Antigens and Immunotherapy: Where Do We Stand in the Targeting of PRAME?"

_cancers, 2019, doi:10.3390/cancers11070984_

Round 1

Reviewer 1 Report

The review is almost all related to PRAME; only a few lines relate to CTA's in general (lines 56-64). Therefore the Title may be misleading and could instead focus on "PRAME as a targetable CTA."

Perhaps another few lines could be devoted to non immunologic approaches (eg inhibitors) to targeting PRAME and why this is not possible or why it may not work  in some cancers

Two other interesting works that might be discussed include the paper by Gerber, JM  (2011) which describes the expression of PRAME in stem cells, which is relevant to the use of PRAME as a target. Also the paper by Mathias, M in Leukemia, (2017) which combines an anti-PRAME antibody with a CD47 blocker.

Lines 100-102  "This section may ..."  seems to be an instruction left in by mistake.

Table 1 is all cellular therapies, but is introduced in a paragraph that also mentions vaccines. Maybe retitle to "Clinical trials of cellular therapies..."

Author Response

1. The review is almost all related to PRAME; only a few lines relate to CTA's in general (lines 56-64). Therefore the Title may be misleading and could instead focus on "PRAME as a targetable CTA."

Answer: We appreciate the reviewer’s suggestion; however, we would prefer not to change the title as we believe the current title will draw more attention and is self-explanatory.

2. Perhaps another few lines could be devoted to non immunologic approaches (eg inhibitors) to targeting PRAME and why this is not possible or why it may not work  in some cancers

Answer: We have added some lines on the limited benefit of using inhibitors and gene silencing to target PRAME (lines 146-148).

3. Two other interesting works that might be discussed include the paper by Gerber, JM  (2011) which describes the expression of PRAME in stem cells, which is relevant to the use of PRAME as a target. Also the paper by Mathias, M in Leukemia, (2017) which combines an anti-PRAME antibody with a CD47 blocker.

Answer: We thank the reviewer for the useful suggestions and have incorporated both references in the revised manuscript (lines 125, 278 and 363-365).

4. Lines 100-102  "This section may ..."  seems to be an instruction left in by mistake.

Answer: We apologize for this oversight and have removed the lines in the revised manuscript.

5. Table 1 is all cellular therapies, but is introduced in a paragraph that also mentions vaccines. Maybe retitle to "Clinical trials of cellular therapies..."

Answer: we thank the reviewer for the suggestion and have changed the title of the table (line 159) and its reference in the manuscript (line 156).

Reviewer 2 Report

 This review provides a well-written and comprehensive overview of literature that does not show overlap with a recently published review. So it certainly would be of value for the readership of Cancers.

Abstract: I would consider to delete the sentence in line 17-18 as this is the main goal of reviews.

 Line 70: Consider the title as PRAME expression in healthy tissue and cancer, as you first describe the regulation in healthy development, and not specific for the cancer types.

Line 100-102: There is a remark in from the authors in the text. I actually agree with the suggestion 

For the paragraph starting at line 104: Upregulation is observed in different cancer types. Do you think there is an overlapping mechanism involved or is it tumor specific, related to the function?

Line 127: What was the rationale for the hypothesis for this study?

Line 130-131: rephrase, now 2 times LRR meaning the same thing

Line 143: In the end of the previous paragraph you talk about studies that show favorable outcomes as well as studies that show unfavorable associations with PRAME expression. The discrepancy could be better explained, concerning differences in study design, confounding factors, etc? How would the beneficial effect of PRAME work?

Line 176-177: There are a couple of statements in the manuscript that are not restricted to PRAME. I would make that clear. Now it seems like these problems may arise specifically for PRAME.

Line 193: this paragraph has a sudden end. I would propose a better bridge towards potential efficacy explained in the next paragraph

Line 274: add ref 78

Line 277 and the figure: This is true for TAA in general and not only PRAME. Make this clear for the readers, especially in the legend of the figure that these go beyond PRAME.

Line 321: Could the authors provide an explanation for the discrepant effects?

Line 367 (conclusions): The authors should also put forward in the conclusion the critical issues that need to be solved for PRAME-specific therapies to be successful. 

Author Response

We thank the reviewer for his/her positive evaluation of our manuscript and we hope it will be of value for our peers in the field and readers of Cancers.

1. Abstract:  I would consider to delete the sentence in line 17-18 as this is the main goal of                reviews.

 Answer: We appreciate the reviewer’s comment and have slightly changed the abstract to reflect the suggestion (lines 16-17).

2. Line 70: Consider the title as PRAME expression in healthy tissue and cancer, as you first describe the regulation in healthy development, and not specific for the cancer types.

 Answer: We have changed the title accordingly (line 70).

3. Line 100-102: There is a remark in from the authors in the text. I actually agree with the               suggestion

Answer: We apologize for this oversight and have removed the lines in the revised manuscript and modified the manuscript accordingly.

4. For the paragraph starting at line 104: Upregulation is observed in different cancer types. Do you think there is an overlapping mechanism involved or is it tumor specific, related to the function?

 Answer: We thank the reviewer for the comment and have now included that PRAME                    function and upregulation may differ depending on tissue-specificity and subcellular localization of PRAME (line 104).

5. Line 127: What was the rationale for the hypothesis for this study?

 Answer: We clarified that since cell motility is an important step in acquiring metastatic                abilities, we investigated the effect of PRAME expression on cancer cell motility (line 127).

6. Line 130-131: rephrase, now 2 times LRR meaning the same thing

Answer: We have rephrased lines 131-132 as requested.

7. Line 143: In the end of the previous paragraph you talk about studies that show favorable outcomes as well as studies that show unfavorable associations with PRAME expression. The discrepancy could be better explained, concerning differences in study design, confounding factors, etc? How would the beneficial effect of PRAME work?

Answer: We believe that the controversial association of PRAME with favorable prognosis in some malignancies might be linked to PRAME-mediated regulation of anti-tumor immunity. We have added some information on the regulation of PD-L1 expression, and on PRAME and PD-L1 expression in leukemic cells which suggests that blast cells with higher PRAME and lower PD-L1 expression are associated with a better clinical outcome (lines 338-340). However, how PRAME and PD-L1 are associated needs to be further investigated.    

8. Line 176-177: There are a couple of statements in the manuscript that are not restricted to PRAME. I would make that clear. Now it seems like these problems may arise specifically for PRAME.

Answer: We thank the reviewer for this very important observation and have adjusted the manuscript throughout (lines 177-179, 255-258, 279-282, 306, 312).  

9. Line 193: this paragraph has a sudden end. I would propose a better bridge towards potential efficacy explained in the next paragraph

Answer: We agree with the reviewer and have added a sentence to bridge both paragraphs (lines 197-198).

10. Line 274: add ref 78

Answer: We have added the corresponding references by Pankov et al, Chang et al and Mathias et al (line 278).

11. Line 277 and the figure: This is true for TAA in general and not only PRAME. Make this clear for the readers, especially in the legend of the figure that these go beyond PRAME

Answer: We have modified lines 279-281 and line 285 (figure legend) to indicate that these challenges and countermeasures are valid for all targeted treatments and not only PRAME.

12. Line 321: Could the authors provide an explanation for the discrepant effects?

Answer: We believe that PRAME can influence anti-tumor immunity in multiple ways. On the one hand, PRAME expression is associated with reduced levels of antigen presentation molecules thereby impeding tumor recognition. On the other hand, PRAME positive tumors can display reduced PD-L1 expression, which contradicts PRAME-mediated immune evasion. However, as mentioned in our manuscript (lines 342-348) we cannot exclude the possibility of exhausted pre-existing high-affinity T cells. Indeed, Weber et al reported that vaccination of patients with pre-existing PRAME-specific T cells does not induce T cell expansion to the same extent as vaccination of patients without pre-existing PRAME specific T cells.

13. Line 367 (conclusions): The authors should also put forward in the conclusion the critical issues that need to be solved for PRAME-specific therapies to be successful.

Answer: We have rephrased lines 391-394 in the conclusion to include the specific challenges that need to be addressed for PRAME-specific immunotherapy.